# Reducing Peak Energy Demand among Residents Who Are Not Billed for Their Electricity Consumption: Experimental Evaluation of Behaviour Change Interventions in a University Setting

**DOI:** 10.3390/ijerph18168406

**Published:** 2021-08-09

**Authors:** Bradley S. Jorgensen, Sarah Fumei, Graeme Byrne

**Affiliations:** 1Department of Management, Sport and Tourism, La Trobe University, Bendigo 3552, Australia; 2ClimateWorks Australia, Monash Sustainable Development Institute, Melbourne 3000, Australia; sarah.fumei@climateworksaustralia.org; 3Research Infrastructure, La Trobe University, Bendigo 3552, Australia; g.byrne@latrobe.edu.au

**Keywords:** energy consumption, daily peak consumption, critical peak consumption, feedback, residential students

## Abstract

Behaviour change interventions aiming to reduce household energy consumption are regarded as an effective means to address disparities between demand and supply and reduce emissions. Less recognised is their success in shifting consumers’ energy consumption from peak demand periods to off-peak times of the day. This study reports two experiments that test the effect of feedback and reminder notifications on energy consumption in university halls-of-residence. A quasi-experiment and a randomised controlled experiment were conducted with residential students to evaluate behaviour change interventions aimed at reducing daily peak and critical peak demand, respectively. The results of Experiment One (*n* = 143) demonstrated significant reductions in the energy use of the treatment group relative to the control. On average, the treatment group’s energy use was 12.4 per cent lower than their pre-intervention baseline. In Experiment Two (*n* = 88), normative elements of the intervention were supplemented with a reminder notification prior to the onset of the simulated critical peak demand period. The results showed that, relative to the control condition, the 8-h notification reduced demand by 20% on average with a 12% decrease for the 24-h notification (with 2-h follow-up). These results indicate that peak energy issues can be alleviated using low-cost and easily implemented behaviour change strategies.

## 1. Introduction

Energy conservation is an important issue for most countries simply because the historical and continued exploitation of non-renewable resources such as fossil fuels is having considerable impact upon the environment. Several strategies are available to governments, organisations and households to achieve reductions in energy consumption [1,2,3], and new tariff structures are listed among them [3,4]. However, in instances where consumers are not billed for their metered consumption, the use of financial incentive structures and mechanisms to influence consumption is not possible [5].

Residential halls on college campuses are one example where individuals are thought to have few incentives to reduce their electricity consumption because its cost is not directly linked to how much they pay in rent [6,7]. Other examples reported in the literature include multifamily apartments [8], residential quarters on military bases [9] and tourist accommodation [10]. In these types of residential settings, energy reductions need to be motivated largely via non-price incentives.

In the following sections of this article, previous research on motivating energy savings in college halls of residents is reviewed. Section 3 discusses different types of peak consumption and the challenge they present for behaviour change interventions. Section 4 describes the context in which this study was undertaken, while subsequent sections of the article detail the methodology of testing two interventions that targeted daily peak and critical peak energy consumption. The results of these trials are then reported along with a discussion of their significance for energy demand management. The article ends with a recognition of study limitations and some general conclusions.

## 2. Review of Energy-Saving Interventions in University Residential Halls

There is a small number of controlled experiments in the peer-reviewed literature that have sought to evaluate the effectiveness of energy-saving interventions in residential halls. All these studies have targeted general reductions in daily energy use rather than peak times of the day or week. Nonetheless, this literature does target similar populations with respect to age, gender, marital status, education level and energy-use behaviours. These parameters in student populations are clearly different to those in the general public [8], and to military [9] and tourist [10] populations where energy billing does not occur. Moreover, the resident composition (e.g., number of residents) in residential halls is dramatically different to the single-family households often studied in the resource conservation literature [2].

All the studies trialled interventions that comprised a mix of behaviour change strategies. These strategies included personal feedback about the consumption of all students in the bedroom [11,12,13,14]; the consumption of the whole building [6,15,16]; descriptive normative feedback concerning the consumption of similar residents and/or efficient residents [11,14,17]; injunctive normative feedback [11,13,14]; persuasive communications [6,11,13,14,15,16]; monetary incentives [15]; non-financial incentives [6,14]; and energy saving education [6,11,13,14,15,16].

The results of the research noted above have shown that significant energy reductions can be achieved in this study population. Only one experiment reported non-significant results [12], and the authors argue that behaviour change interventions require monetary incentives tied directly to energy consumption to achieve their greatest impact. The experiment by Myers and Souza did not analyse consumption data directly, but targeted thermostat use in the students’ bedrooms. The dependent variable was the difference between the thermostat settings and outdoor temperature. The researchers compared the effect of their intervention comprising weekly emails reporting students’ own energy consumption, normative information about participants including information about their energy efficiency ranking, and recommendations for adjusting their thermostats. The thermostat behaviour of the intervention was monitored for rooms and suites (i.e., groups of rooms) and compared with a control. No significant effect was observed at either the level of the room or the suite.

Other controlled trials in college dormitories have reported significant energy savings. McClelland and Belsten [15] achieved a reduction equivalent to 84 per cent of baseline consumption (i.e., 50,000 kwh or 3.85 kwh/resident/day) with an intervention comprising persuasive communications, education, feedback, and group meetings implemented over 10 weeks. A second 6-week intervention added a monetary reward to the mix of behaviour change strategies and resulted in consumption levels equal to 90 per cent of the new baseline established after the savings achieved by the first intervention (i.e., 17,000 kwh or 3.44 kwh/resident/day).

Even quite short interventions can result in energy savings. In just a 2-week program, Petersen et al. [16] compared two levels of feedback: high resolution, real-time feedback and low resolution, weekly feedback. The intervention overall resulted in an average consumption level of 32 per cent of baseline (or 68,300 kWh). However, greater savings were achieved in the high resolution condition, which experienced a 55 per cent reduction versus 31 per cent in the weekly feedback condition. Bekker et al. [6] also trialled a short, 3-week intervention comprising education, persuasive communications, feedback (provided each week day), and non-financial rewards (e.g., free coffee) over 3 weeks to achieve a relatively modest 3700.31 kWh reduction in consumption (or 13.45 per cent of baseline).

Of the studies that measured consumption at the bedroom level, significant reductions were achieved from interventions that combined three or more of the following strategies: personal feedback, normative feedback, persuasive communications, energy-saving tips, and non-financial rewards. Delmas and Lessem [13] observed energy savings of 20 per cent of mean consumption over the trial period, and Alberts et al. [14] reported 22 per cent savings compared to consumption in the control condition.

Anderson et al. [11] provide evidence that intervention effects can vary with population subgroups that differ with respect to factors likely to be responsive to particular components of the intervention. They report that participants in their experiment who had a high concern for social norms reduced their energy consumption by 14 per cent of baseline levels. Individuals with a low concern for social norms, on the other hand, increased their consumption by 5 percent. Furthermore, the long-term effect of behaviour change was twice as prevalent in individuals with high concern for social norms.

These results for student residential studies are consistent with the broader literature concerning energy use in households [18,19] and somewhat more overlapping domains such as hotel accommodation [10] and military installations [9]. Moreover, like the interventions trialled in these other contexts, most of the experimental work in college halls combines different types of behavioural feedback (i.e., individualised performance information versus descriptive normative feedback) into a single behaviour-change program. More often than not, behavioural feedback is also supplemented with educational information (e.g., conservation tips), reminder notifications, and/or injunctive normative messaging. This confounding is understandable from at least two perspectives. First, a review of the residential energy and resource use literature has shown that feedback interventions work better when combined with other interventions [2]. Second, while the trials conducted in halls have clear theoretical objectives, they also have an applied goal of achieving meaningful levels of demand reduction that will benefit both the environment and the university. Therefore, interventions are justifiably given the greatest chance of success.

One major point of variation across the controlled trials implemented in college residential halls has been their level of analysis, namely, the individual, the bedroom group, and the entire residential hall. However, even when the bedroom is the point at which theory is applied and data collected, none of the studies appear to involve purely individual-level energy consumption. For example, Anderson et al. [11] reported 276 and 219 student participants housed in 220 and 152 rooms, respectively. Similarly, the experiment by Myer and Souza [12] involved 330 bedrooms housing 400 students and did not analyse energy consumption. Not all bedrooms were single occupancy such that the dependent variable in these experiments reflects the consumption of one or more individuals. Therefore, the effect of the interventions in these experiments may have been diminished due to a misalignment between the unit of analysis targeted by the intervention and the measurement of the dependent variable [20].

## 3. A Focus on Peak Energy Demand

The interventions employed in previous research target energy savings from any part of the day and, therefore, are indiscriminate with respect to the high points of consumption in the daily energy demand distribution [21,22]. This distribution is known to peak during certain hours of the day (in households at least) and this can create episodes of high demand that threaten daily supply (referred to as “daily peak demand”). In contrast, “critical peak demand” occurs on a relatively small number of days in the year where demand is at extremely high levels and threatens the reliability of supply. Peaks like these in household energy demand can be “flattened” using behavioural interventions and monetary incentives [23], although evidence has also been mixed [18,24]. Nonetheless, little is known about the effectiveness of behavioural interventions in university settings.

Flattening out daily peak and critical peak demand can improve the cost-effectiveness of the energy infrastructure but requires consumers to alter their behaviour to achieve it [24]. To this end, retail energy companies in Australia and elsewhere have introduced a range of “energy-based pricing” that provides financial incentives to lower energy use on very high demand days of the year and/or time of the day [25]. Importantly, the goal of strategies aimed at “smoothing” the consumption distribution is not to realise energy savings but to decrease energy use at specific times of the day or week. To achieve this, individuals do not need to reduce their overall energy consumption because they can simply shift their energy use to another off-peak time. This distinction notwithstanding, a focus on peak energy use can involve absolute reductions in energy consumption, but these must be achieved during the peak period(s).

## 4. The Study Context

This study forms part of Monash University’s Net Zero Initiative, which commits it to achieving net zero emissions on its Australian campuses by 2030. The University has invested in renewable energy generation, building a microgrid at its main campus in Melbourne, Australia. Through reducing peak demand, Monash can use more of the renewable energy generated on campus, rely less on electricity purchased off-site, and reduce overall costs.

The current research conducted experimental trials in two halls of residence, which together house 600 students in multi-unit, single-occupant accommodation. The halls provide an ideal setting for a demand management experiment because students do not share rooms, the single-occupant apartments are all almost identical in size, and each dwelling is individually metered. Students have control over their operation of lighting, cooking equipment, electric heaters and fans, entertainment equipment, computers and other appliances. Fridges, hotplates and ceiling lights are provided by the University. Gas heating is provided centrally and there are no air conditioners installed in any of the apartments.

Following McClelland and Belsten [15], two behaviour change interventions were tested during 2018. In Semester 1, normative feedback delivered weekly for five weeks was trialled using a quasi-experimental design. This tested the following hypothesis:

**Hypothesis** **1** **(H1).***The provision of normative feedback enabling social comparisons would result in a greater reduction in daily peak energy use than the provision of information about energy saving tips*.

The second intervention was tested in Semester 2 using a randomised design to evaluate the following hypothesis: In conjunction with the provision of normative feedback, the notification time prior to the onset of the target peak period (i.e., no notification, an 8-h notification, and a two-time notification at 24-h and 2 h prior to the peak time onset) influences the amount of energy reduction. A more specific hypothesis followed from the work of Luyben [26], who identified the timing and frequency of the prompt as potentially influential for behaviour. In this respect, one might expect the 24 + 2-h notification to be most effective because of its greater frequency and proximity to behaviour.

## 5. Materials and Methods

### 5.1. Experiment One

#### 5.1.1. Participants

All 600 students in the residential halls were sent an email from Monash Residential Services inviting them to participate. The email explained that reducing peak energy demand has important benefits for the grid, the University, and for the environment, and that by participating in the study students would be in the running to win a AUD 200 gift voucher. In total, 143 students registered to participate in the study (24 per cent sign up rate). All participants were full-time students, more than 90% were aged between 18 and 24 years, and the majority of participants (85%) had completed up to three years of study. There was slightly higher representation of females than males (57 per cent and 43 per cent, respectively) in keeping with the profile of the University student population.

Students in one residential building were selected as the control group (*n* = 70), while the residents of the second building were allocated to the treatment group (*n* = 73). The survey data were analysed to ensure balance between the control group and the treatment group. The analysis showed that the control and treatment groups were not significantly different on age, year of study, full-time study, and gender (*p* > 0.05) (see Table 1). Moreover, the areas of study profiles for control and treatment groups were not significantly different, with a *z*-test of equality of proportions between groups for each area of study providing null results in all cases (see Figure 1).

#### 5.1.2. Procedures

Students registered for the study online where they were informed about the purpose of the study and were required to complete a survey. Students were informed that the peak time for the residential halls was 8 p.m. to 10 p.m. on weeknights and included questions about students’ age; area of study; behavioural intentions to save energy; environmental concern [27]; and environmental indifference (see Table 2). Key variables in Schwartz’s [28] Norm Activation Model of prosocial behaviour and Stern et al.’s Value-Belief-Norm Theory [29] of pro-environmental behaviour (i.e., feelings of moral obligation to reduce energy consumption or “Personal Norm”; level of acceptance of responsibility for reducing energy demand; and awareness of the consequences of energy use) were also measured. Students’ individual energy use was monitored for both the treatment and the control group. There was an initial 2-week period of data collection, which provided an energy use baseline against which subsequent energy use could be compared.

The treatment group received weekly emails consisting of the following six components:A request that the students reduce their energy use during peak times (8 p.m. to 10 p.m. on weeknights) [6,13,15,16].A reminder that reducing energy use during peak times has important benefits for the environment and the University [26,30].A comparison of the student’s energy use with that of an efficient student and that of an average student [11,12,14].The student’s ranking in comparison to other students in the treatment group—the student with the lowest peak demand for the week was ranked as #1, while the student with the highest peak demand was ranked as #73.A link to a dashboard with more detailed information about the student’s energy use, including a 24-h demand profile, and a chart tracking their progress across the weeks of the study [13].Tips for reducing energy use, which were updated each week [5,13,14,15].

More information about the intervention described above and the study methodology can be found in ClimateWorks Australia [31].

### 5.2. Experiment Two

#### 5.2.1. Participants

The 600 students in the same residential halls as the first trial were sent an email inviting them to participate in a second experiment. The email explained that by participating in the trial they would be in the running to win a AUD 200 gift voucher, and that there would be a survey at the end of the trial that would provide them with another chance to win a AUD 200 gift voucher. Students registered for the study online where they provided information about their age, area of study, preference for email or SMS notifications or both, as well as consent to monitor and analyse their energy consumption data. In total, 88 students completed the survey and registered to participate in the study (15 per cent participation rate).

#### 5.2.2. Procedures

Eight simulated critical peak demand events (from 8 p.m. to 10 p.m.) took place on weeknights over the semester. The trial aimed to test the effect of different notification periods for peak demand events on electricity consumption. There were three notification periods prior to the peak events:No notification (control condition).An 8-h notification.A 24-h notification with a 2-h reminder (24 + 2 condition).

A fully counterbalanced, within-subjects randomised experimental design was developed to account for potential changes in students’ energy use depending on the order in which they received the different treatment levels [32]. In this way, all participants received all three notification periods during the trial and served as their own control. An initial 4-week period of data collection provided a baseline against which subsequent energy use could be compared.

For each critical peak demand event, all students who received a notification reminder also received the following information:Tips for reducing energy use, which were updated each week [6,14,15,16].Normative messaging:
-Students who reduced their energy use in comparison to their baseline received: “Good work! To save even more energy try following the tips below during the next peak demand event.” [14]-Students who increased their energy use in comparison to their baseline received: “Looks like you’re having a bit of trouble saving energy-the tips below might help you during the next peak demand event.”
The percentage difference in the student’s energy use in comparison to their baseline.A chart showing their baseline energy demand during peak times in comparison to their energy demand during the simulated peak demand event.The student’s ranking in comparison to other students.

Students who were in the control condition for a peak demand event received their percentage change in energy use, the chart, and the tips for reducing energy use, but did not receive a ranking or normative feedback.

Following the 8-week trial, students were invited to participate in a survey. A total of 62 students participated in the survey (70 per cent of participants) for which they received a chance to win a AUD 200 gift voucher. The survey aimed to provide a better understanding of students’ experiences during the trial. It included questions on participants’ preferences for an 8-h notification or 24-h notification with a 2-h reminder, and their preference for a device that automatically switches off energy-using appliances.

## 6. Results and Discussion

### 6.1. Experiment One Results and Discussion

#### 6.1.1. Survey Analysis

Prior to testing the effect of the intervention on energy consumption, the treatment and control groups were compared to identify any differences in the social psychological variables (i.e., behavioural intentions to save energy; understanding of responsibility for energy demand; awareness of the consequences of energy use; environmental concern; and environmental indifference) theoretically related to the behaviour. A series of Mann–Whitney U-tests (Table 3) found no significant differences between the treatment and control groups for the social psychological variables. Therefore, pre-trial differences in these variables could be ruled out as possible explanation of any treatment effect.

#### 6.1.2. Energy Consumption Analysis

A mixed model repeated measure (MMRM) analysis of covariance was conducted to analyse the mean energy use data gathered from both the treatment and control groups [33]. This analysis was carried out using a MMRM with an unstructured covariance type to allow for possible time dependence of observations within each subject [34]. The time-period and group were set as fixed factors, while the six social psychological variables were defined as covariates. The MMRM found a significant group-by-time interaction (*p* = 0.043), suggesting the mean energy use of both groups is not parallel, as represented in Figure 2. Additionally, the test demonstrated that, out of the six psychological variables surveyed, environmental indifference was the only significant covariate (*p* = 0.007), with higher energy use associated with higher levels of environmental indifference (see Table 4).

Having identified that the treatment and control groups showed different patterns of energy use over time, the data analysis then focused on studying the differences between the pre-treatment period and each post-treatment period. The results presented in Table 5 show that, for the control group, the five differences (between the pre-trial period and each week of the study) were identified as not significant (all *p*-values > 0.05). In contrast, the treatment group reduced their energy use after the first email, and maintained this behaviour, with only one exception, during the whole study period. Significant reductions were identified in the energy use of the treatment group (*p* < 0.05) in all post-treatment periods except for the period between 3 May 2018 and 7 May 2018 (*p* = 0.135). Averaged across the entire study, we found that for the treatment group, energy use was 12.4 per cent lower than the baseline.

Further investigation of the data for the period between 3 May 2018 and 7 May 2018 did not provide a clear reason for the non-significant result in this week. We did not find any data errors and there were no particularly hot or cold days during the period in question. The data for the control group demonstrate that in the absence of intervention, peak demand for students is inherently variable (see Figure 2). The week when the treatment group did not see a statistically significant decrease in peak demand is likely a result of this natural variability.

Experiment One showed that the combination of personalised behavioural feedback and normative feedback can serve as effective components of demand management programs. Participants in the treatment group received a simple non-financial, feedback-based intervention and reduced their daily peak demand by an average of 12.4 per cent in comparison to their baseline energy use.

Comparison of the effectiveness of the intervention with past research is not straightforward given the focus here on daily peak demand rather than on overall demand reduction. However, we note that the 12.4 per cent reduction in daily peak demand is consistent with results in the general literature conducted in residential halls, which vary between zero and 55% (e.g., [6,11,12,14,15,16,17]). One outlier is the results of Petersen et al. [16], who reported energy reductions of between 31% and 55% for their low- and high-resolution feedback conditions, respectively. Unlike most other studies in the literature, the intervention lasted just two weeks and was presented as a competition between dormitories with the chance of winning an invitation to an ice cream party. Perhaps students engaged in reducing energy consumption in the context of an intensive competition can achieve relatively significant energy reductions over the short term. Furthermore, Petersen et al. showed that the energy savings achieved by their intervention persisted approximately four weeks after its completion. More research is required to better understand how the effect of normative feedback on peak consumption can be enhanced by gamification techniques. Using game-oriented behavioural interventions, cooperative and competitive motivations and their interactions can be compared with normative motivations directed toward reducing peak consumption [35,36].

The intervention effect in Experiment One was observed after controlling for pro-environmental motivations according to the Norm Activation Model [28] and the New Ecological Paradigm [27]. Of these motivations, only the Environmental Indifference factor had a significant effect on energy consumption over and above the effect of the intervention. Therefore, the intervention was not effective in reducing the energy consumption of participants who believed in the virtually unbounded capacity of humankind to overcome a supposedly “over-stated” ecological crisis. Additional studies are required that can show exactly what motivational strategies are effective in influencing the peak consumption of various population subgroups, especially those groups that are difficult to engage [37,38].

### 6.2. Experiment Two Results and Discussion

#### 6.2.1. Statistical Model

The statistical model used in the analysis was based on Kuehl (2000) and took the following form:(1)$$log\;(Y_{ijkl})=\mu +\alpha_{i}+b_{j\left(i\right)}+\gamma_{k}+c+\lambda_{d\left(i,k-1\right)}+\delta_{l}+\epsilon_{ijkl}$$
where$Y_{ijkl}$ is the response due to subject i, treatment j, period k and sequence l.μ is the overall mean.$\alpha_{i}$ is the fixed effect of the ith sequence.$b_{j\left(i\right)}$ is the random effect of subject i nested within sequence j.$\gamma_{k}$ is the fixed effect due to period k.$\tau_{c\left(i,k\right)}$ is the direct effect of treatment administered in period k of sequence i.$\lambda_{d\left(i,k-1\right)}$ is the carryover effect of the treatment administered in period k−1 of sequence i.$\delta_{l}\;$is the fixed blocking effect to account for variation between buildings.$\epsilon_{ijkl}$ is the random error term.

A robust estimation procedure was used to estimate the model parameters, which avoided the need for meeting the normality and constant variance assumptions [34].

#### 6.2.2. Energy Use Analysis

A generalised linear mixed model using a log-link function was employed to test the model. The time period (i.e., the week of the intervention), sequence of events (i.e., the counter-balancing of the treatment levels in the design), residential hall (i.e., which of the two residential halls the student resided in) and treatment (i.e., whether students received no notification, 8-h notification or 24-h notification with a 2-h reminder) were set as fixed factors.

It was found that the sequence of events (*p* = 0.025), the time period (*p* = 0.001) and the treatment (*p* = 0.001) all had a significant effect on students’ energy use, while residential hall did not (*p* = 0.237). Therefore, after controlling for differences due to the order of presentation of the notifications and the week of the semester, the type of notification had a statistically significant effect on peak consumption.

The impact of the treatment was analysed further to better understand the effect on mean energy savings for the 8-h notification in comparison to the 24-h notification with a 2-h reminder. For the control, average peak demand was 1.85 kW, while for the 8-h notification, it was 1.47 kW (20% decrease), and for the 24-h notification (with 2-h reminder) it was 1.62 kW (12% decrease).

Table 6 shows the results of comparing the reminder treatment levels to the control. While there was a significant difference between the 8-h notification and the control (*p* = 0.003), the difference between the 24-h notification with a 2-h reminder and the control was borderline significant (*p* = 0.058). This implies that the 8-h notification is an effective way to promote a reduction in energy use, while the 24-h notification with a 2-h reminder had a marginal influence on peak consumption relative to the control.

#### 6.2.3. Ease of Reducing Energy Consumption during Peak Time

There were mixed evaluations of how easy the intervention was to comply with. Approximately 42% of respondents said it was somewhat or extremely easy to reduce their energy use, while around 32% said it was somewhat or extremely difficult. The major reasons given for any difficulty in reducing energy use were cooking (27.4%), study (24.2%), and computer use or charging (12.9%). The major reasons given for making it easy to reduce energy use were not being in the room (25.8%), not needing heating or cooling (8.1%) and use of communal study areas (8.1%).

#### 6.2.4. Participant Preferences for Different Notification Periods

Participants were asked about their preferences for different notification periods about upcoming peak demand events. While 45 per cent of respondents had no preference for a particular notification period, 39 per cent preferred the 24-h notification with a 2-h reminder, while 16 per cent preferred an 8-h notification. Respondents who preferred the 24-h notification with a 2-h reminder said that it provided a better ability to plan, and that they found multiple reminders helpful. Respondents who preferred the 8-h notification said that they preferred not to receive multiple emails.

#### 6.2.5. Participant Preferences for Automatic versus Manual Control of Appliances

Participants were asked about their preferences for switching appliances off themselves or having a Wi-Fi enabled plug that would be able to switch off appliances automatically. Almost half the respondents (48 per cent) expressed a preference for manually switching off appliances, while 24 per cent expressed a preference for automation, and 27 per cent had no preference for either method. Those who preferred the manual option said that they wanted to maintain control, and that they were concerned that the plug would switch off appliances that were needed or in use. They were also concerned that technical errors would make the plug less effective. Those who preferred the automated option said that it would be more convenient and would ensure they did not forget about an upcoming peak demand event.

These results demonstrate that convenience and control were important considerations for participants. Achieving this mix of making the behaviour easy but allowing the individual to retain control of their environment and behaviour is potentially a challenging goal. One way of achieving it, however, would be to match the energy saving strategy with each individual’s preference. Including members of the target population in the design of interventions, especially those where individual control might be taken away, would assist in developing demand management strategies that recognise the role of both convenience and control.

A hypothesis tested in Experiment Two was whether the timing of peak demand notifications influenced participants’ responses to peak demand events above and beyond the normative feedback effects demonstrated in Experiment One. Participants who received an 8-h notification of an upcoming peak demand event reduced their peak demand by 20 per cent in comparison to those who did not receive a notification. This result was counter to expectations where the 24 + 2 condition was hypothesised to have a greater influence on consumption because of the added frequency of the presentation of the prompt and its greater temporal proximity to behaviour.

It is unclear as to why the 8-h reminder was more effective, except to speculate that the additional frequency of the 24 + 2 prompt might have provoked psychological reactance. Reactance is a process whereby individuals perceive a threat to their autonomy and act in opposition to the behaviour change appeal [39]. Lehner et al. [40] noted that, while prompts can be low-cost interventions, they can be experienced negatively by some consumers (see also [41,42]). This possibility notwithstanding, of those participants who registered a preference for a notification period, the majority (71 percent) favoured the 24 + 2 option, suggesting an absence of reactance among this group. Nonetheless, reactance may have been a factor for the other 29 percent of participants who preferred the 8-h period and the 45 percent of the sample who did not express a preference. Irrespective of what underlying mechanism might explain the result, students’ preference for a particular reminder does not directly correspond with the most effective reminder in terms of reducing energy consumption. In this sense, tailoring the timing of the prompt to satisfy individual preferences might not produce the desired behaviour change.

The timing of peak demand notifications has not previously been experimentally tested to our knowledge. Reminder prompts presumably work best when most proximal to the presentation of the opportunity to act; however, little is known about the optimal timing of such notifications for behaviour change or the ideal conditions of use generally [43]. Research on the influence of prompts usually examines potential effects in terms of their presence or absence in an intervention rather than their timing [30,44,45]. This research gap stands despite one study demonstrating that the timing of the prompt is more important than its frequency of presentation [26].

Previous studies on critical peak demand management have used varying notification periods (e.g., [46,47]) but have not included their temporal variation as an independent variable. Future research would benefit from designs that test a larger frequency of prompts than the combination of just two prompts we employed in our 24 + 2 condition. It seems reasonable that some participants might find repeated prompting unnecessary and annoying, leading to reactance. Alternatively, participants might experience greater frustration with an increasing number of reminder prompts, but still seek to reduce their energy consumption. In addition to the temporal characteristics of reminder prompts, the mode by which the reminder is provided may prove influential, especially if delivered face-to-face and/or by a valued peer rather than electronically and/or anonymously. Further investigation of these aspects of reminder prompts, rather than simply the absence or presence of a prompt, could assist in developing an identifying the conditions that are most effective in alleviating peak demand.

The survey responses provided by participants in Experiment Two highlight several factors that might be applied in college settings to manage critical peak energy events. Providing some support for student efforts to reduce their energy consumption to flatten peak demand is necessary given that one-third of participants reported some level of difficulty in their efforts to achieve it. The need to prepare meals and study during times of peak demand were two main barriers to behaviour. Making target behaviours easy to enact is basic to achieving desired behaviour change [48,49]. Enabling students to prepare meals and study outside periods of critical peak demand would assist in addressing these concerns. Additional strategies might be to make pre-prepared meals available to students, and to encourage group study options at peak energy times.

Participants noted that the activity that facilitated their change of energy use behaviour was simply not being in their rooms during peak events. One problem this presents for energy management is that a conservation strategy may be ineffective if students’ energy use continues somewhere else on campus. Demand management communications need to target reducing energy consumption *anywhere on campus* during peak events. This insight notwithstanding, our data do not enable us to know where students relocated to and what their energy use might have been for those students who vacated their rooms but remained on campus.

In summary, this study focused on the effectiveness of feedback and reminder prompts as a motivational strategy for reducing daily peak demand and simulated critical peak demand. The general conclusion that can be drawn is that relatively simple and low-cost behaviour change strategies can achieve significant success in managing both daily peak and critical peak occurrences. As has been demonstrated by previous studies involving university students [11,16], these behaviour changes can be achieved without the need for billing. These findings advance the body of knowledge on behavioural research by applying normative concepts to the promotion of energy conservation. The results also demonstrate that reminder prompts can assist in shifting consumption patterns from peak periods to off-peak times.

The improvements in managing electricity grids by reducing peaks in consumption mean that additional infrastructure investment to increase capacity can be either delayed or avoided. Moreover, with reductions in peak demand, electricity is made available for other high priority services. We discuss the significance of the results for demand management in the following sections.

### 6.3. Study Limitations

A limitation of our experimental designs was the combination of different forms of feedback in a single intervention in Experiment One and conflating normative feedback with the timing of the prompt in Experiment Two. These design limitations were due to sample size constraints and the need to maximise statistical power. Future research in contexts with access to larger populations of potential participants would assist in recruiting samples large enough to enable the study and compare the unique effects of these behaviour change strategies and the psychological mechanisms underpinning them.

Although both interventions proved successful in helping participants reduce their energy consumption in their rooms, we do not have data to indicate whether they continued to use energy somewhere else on campus. Therefore, the second limitation of this and all energy demand management research is the lack of information about the consumption behaviour of participants when they leave their residence. Simply not being resident reduces energy consumption when it is monitored at a specific site such as a room in a residential hall, household or business.

## 7. Conclusions

Two experiments were conducted with the participation of student residents in university halls on campus. The results of Experiment One showed that the provision of normative feedback significantly reduced daily peak energy use compared with the provision of information about energy saving tips. Experiment Two generated evidence that the notification of normative feedback prior to the onset of a target peak period influences the amount of energy saved. A reminder 8 h prior to the critical peak period had a greater influence on energy reduction than a two-time notification at 24 h and 2 h prior to the peak and no notification at all.

To our knowledge, this study is the first to investigate the effectiveness of behaviour change interventions targeting peak demand in college residential halls. Moreover, the results suggest that energy savings can be achieved that are comparable to those achieved by interventions targeting overall energy reductions in similar settings. In addition to any absolute energy savings that universities may reap from behavioural interventions, there is the added efficiency and infrastructure benefits associated with improvements in the management of the electricity grid. Furthermore, the types of behavioural interventions trialled here have relatively low implementation costs and can be routinely carried out each semester or as required.

## Figures and Tables

**Figure 1 ijerph-18-08406-f001:**
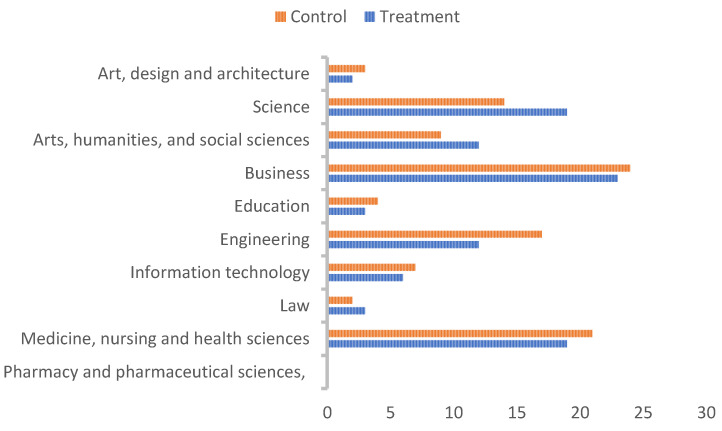
Area of study by group (%).

**Figure 2 ijerph-18-08406-f002:**
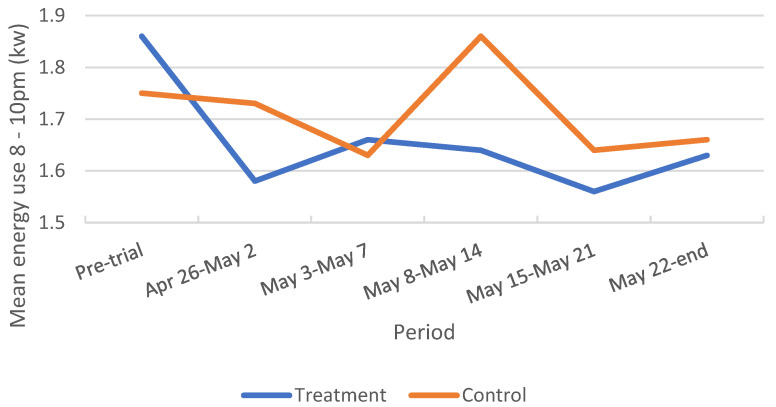
Mean energy use between 8 p.m. and 10 p.m. by group.

**Table 1 ijerph-18-08406-t001:** Comparison of descriptive statistics by group.

Variable		Control %	Treatment %
Age	18–24	47	53
	25–34	33	67
Full-time study	Full-time	45	55
Years of study completed	One	41	59
	Two	65	35
	Three	39	61
	Four	29	71
	Five or more	33	67
Gender	Male	48	52
	Female	43	57

**Table 2 ijerph-18-08406-t002:** Survey questions and reliability.

Variable	Scale Questions	Cronbach’s α
Behavioural Intention (BI)	1. Over this semester, how likely or unlikely are you to reduce your electricity use in your room during a high demand period? 2. Over this semester, how likely or unlikely are you to perform the following behaviours in your room during a high demand period? Turn off my air conditioner Turn my air conditioner “down” (i.e., to a higher temperature setting) Turn off my electric heater Turn my electric heater down Turn off my electric fan Turn my electric fan down Only turn my computer on before or after the high demand period Only use my electric hotplate/oven before or after the high demand period Only use my microwave before or after the high demand period Only use my dishwasher before or after high demand period Turn off at the wall any appliances not being used Turn off all lights not being used Turn off what I can and leave my dormitory room [7-point rating scale from *extremely likely* to *extremely unlikely*.]	0.85
Personal Norm (PN)	3. I am willing to put extra effort into reducing electricity consumption in my room during high demand periods this semester.	0.89
	4. I feel morally obliged to take actions that reduce electricity consumption in my room during high demand periods this semester.	
	5. I believe that I should avoid using electrical appliances in my room during high demand periods this semester.	
	6. I feel personally obligated to reduce the use of electricity during high demand periods this semester.	
	[7-point rating scale from *strongly agree* to *strongly disagree*.]	
Acceptance of Responsibility (AR)	7. Helping to reduce strain on the network during high demand periods this semester by reducing my electricity consumption in my room. 8. Helping to reduce the frequency of power outages caused by high electricity use during high demand periods this semester. 9. Helping to reduce the use of electricity in your room during high demand times this semester. [4-point rating scale from *completely responsible* to *not at all responsible*.]	0.91
Awareness of Consequences (AC)	10. Make more electricity available to provide vital services at the University. 11. Save money for the University. 12. Help postpone the need for the University to undertake costly investment in electricity infrastructure. 13. Reduce carbon pollution in the atmosphere from coal powered electricity generators. 14. Provide an enjoyable challenge. 15. Make me feel good about doing something worthwhile.	0.68
	[5-point rating scale from *definitely true* to *definitely false*.]	
Environmental Concern (EC)	16. We are approaching the limit of the number of people the earth can support. 17. When humans interfere with nature it often produces disastrous consequences. 18. Humans are severely abusing the environment. 19. Plants and animals have as much right as humans to exist. 20. Despite our special abilities, humans are still subject to the laws of nature. 21. The earth is like a spaceship with very limited room and resources. 22. The balance of nature is very delicate and easily upset. 23. If things continue on their present course, we will soon experience a major ecological catastrophe.	0.77
	[5-point rating scale from *strongly agree* to *strongly disagree*.]	
Environmental Indifference (EI)	24. Humans have the right to modify the natural environment to suit their needs. 25. Human ingenuity will ensure that we do NOT make the earth unliveable. 26. The earth has plenty of natural resources if we just learn how to develop them. 27. The balance of nature is strong enough to cope with the impacts of modern industrial nations. 28. The so-called “ecological crisis” facing humankind has been greatly exaggerated. 29. Humans were meant to rule over the rest of nature. 30. Humans will eventually learn enough about how nature works to be able to control it. [5-point rating scale from *strongly agree* to *strongly disagree*.]	0.81

**Table 3 ijerph-18-08406-t003:** Mann–Whitney U-tests of difference between the treatment and control groups for the social psychological variables.

Variable	Mann–Whitney U-Test *p*-Value
Behavioural Intentions (BI)	0.234
Personal Norm (PN)	0.482
Acceptance of Responsibility (AR)	0.234
Awareness of Consequences (AC)	0.430
Environmental Concern (EC)	0.694
Environmental Indifference (EI)	0.438

**Table 4 ijerph-18-08406-t004:** MMRM results of energy use 8–10 p.m. weekdays (kW).

Variable	Numerator df	Denominator df	F	Sig.
Intercept	1	121.214	15.548	0.000
Group	1	121.373	0.235	0.629
Time	5	127.000	4.165	0.002
Group × Time	5	127.000	2.373	0.043
Intentions	1	121.000	1.787	0.184
PN	1	121.000	2.295	0.132
AR	1	121.000	0.359	0.550
AC	1	121.000	0.383	0.537
EC	1	121.000	2.664	0.105
EI	1	121.000	7.598	0.007

**Table 5 ijerph-18-08406-t005:** Simple period effects within treatment and control groups.

Group	Period	Mean Difference	Std. Error	df	*p*	95% CI	
						Lower Bound	Upper Bound
Control	26 April–2 May	−0.045	0.068	127	0.972	−0.223	0.133
	3 May–7 May	−0.100	0.094	127	0.822	−0.345	0.146
	8 May–14 May	0.113	0.088	127	0.679	−0.117	0.343
	15 May–21 May	−0.133	0.088	127	0.512	−0.362	0.097
	21 May–end	−0.099	0.085	127	0.755	−0.321	0.122
Treatment	26 April–2 May	−0.269	0.060	127	0.000	−0.424	−0.113
	3 May–7 May	−0.183	0.083	127	0.135	−0.398	0.032
	8 May–14 May	−0.211	0.077	127	0.036	−0.413	−0.009
	15 May–21 May	−0.286	0.077	127	0.002	−0.486	−0.085
	21 May–end	−0.220	0.074	127	0.019	−0.414	−0.026

**Table 6 ijerph-18-08406-t006:** Individual reminder treatment contrast results.

Treatment Simple Contrast	Contrast Estimate	Std. Error	Bonferroni Sig.
8-h notification email vs. Control	−0.375	0.119	0.003
24 + 2-h notification email with reminder vs. Control	−0.277	0.118	0.058

## Data Availability

Restrictions apply to the availability of these data. Data was obtained from United Energy and are available from the authors with the permission of United Energy.

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
