# Peer review of "Reducing Peak Energy Demand among Residents Who Are Not Billed for Their Electricity Consumption: Experimental Evaluation of Behaviour Change Interventions in a University Setting"

_ijerph, 2021, doi:10.3390/ijerph18168406_

Round 1
Reviewer 1 Report
Authors made a study on shifting consumers’ energy consumption from peak demand periods to off-peak times of the day. They applied this study in Monash University’s and by reducing peak demand, Monash can use more of the renewable energy generated on campus, rely less on electricity purchased off-site, and reduce overall costs.
It is a interesting paper and easy to read.
Some questions/suggestions:
1) In Introduction authors must enhanced this paper major contributions and are also encouraged to provide a paragraph refering what is the happening in the rest of the paper.
2) In total, 143 students registered to participate in the study (24 per cent sign up rate). Did authors planned a second approach to increase this value?
Reviewer 2 Report
This paper evaluates the change of people’s behavior who are not obligated for their electricity bill such as university dorm student through proposing two methods namely: through feedback and reminder.
The idea seems interesting and without major initial costs and it was shown that the people are willing to change their consumption behavior so their consumption is moved to off-peak hours.
This paper is interesting from a social science standpoint ( to ask people to shift consumption).
However, since not many people are aware of the power each appliance consumes, the students could be helped by being given the average consumption of each instrument. For example, not using a hair-drier or induction cooktop for 5 minutes consumes more power than a laptop or light uses in one hour. So in short, if people are well-informed about the consumption of appliances, they could effectively reduce even more of their consumption without losing their comfort of using an appliance.
Another issue: the quality of figures is very low and only in grayscale. The quality must be improved.
Reviewer 3 Report
This experimental research reports two experimental tests reporting the effect of feedback and reminder notifications on energy consumption in University halls of residence, aimed at reducing daily peak and critical peak energy demand. These types of notifications will help energy management to reduce energy consumption and can help to define the ways people use energy. This research is a very relevant study focusing on the understanding of strategies to reduce energy consumptions to support the energy transition. This is an interesting topic strongly connected to “IJERPH-MDPI” aims and scopes.
The authors don't refer in the paper abstract the main achievements of this research regarding the effect of this communication strategy on changing behaviors and reducing energy consumption. They just limit themselves to conclude that the main results point out some positive effects by quantifying the results of the investigation however, the authors don't refer in the article abstract the main achievements of this research regarding the strategy to adopt to mitigate peak energy demand. The abstract must include some insights on this topic.
The structure of the article can be improved since there are two missing sections: a framing section to point out the related works on the subject, and results and discussion section where the authors should correlate the effectiveness of feedback and reminder prompts with daily peak demand. Additionally, section 5 (conclusions) should not only establish pieces of evidence based on results but also must frame this study in the context of all related studies that have played an important role in investigating motivational strategies for reducing daily peak demand. A better organization of the paper is strongly needed.
There is a missing section to point out the related works on the subject. Section 2 refers to Materials and Methods. Section 2 must focus on a detailed literature review including related works. The authors didn't include in the research a specific section for “Literature review”. The related works are presented in Section 1 (Introduction). The number of references is short (38) and is developed without relating the quoted studies to each other. It is convenient to reduce the explanatory content of each reference plus research related to the subject of the research. Some new references should be added in a specific Section called “Literature review”, “Related works” or “State of the Art”. All new references should be related to the analyzed subject and it would be appreciated to quote some articles published in the Journal Buildings of MDPI regarding the investigation subject.
In section 1 (Introduction) authors don´t refer adequately to the purpose of the research. The structure of the article is not presented. How many Sections and its name? What is the content presented in each Section? In short, how is the article organized? On the other hand, they refer that they have tried to improve energy peak demand by implementing a communication strategy. How was this improvement measured?
Section 2 refers to Materials and Methods. Figures 2 and 3 must be included in an annex to the paper text. The procedures described in Section 2 must be part of a strategy to improve communication regarding energy peak demand improvement. Therefore, the implemented methods should be better described as part of a strategy to achieve a higher purpose – reduce energy consumption and mitigate CO2 emissions-. A schematic or drawing of the general arrangement of all implemented methodology would be welcome.
Section 3 refers to “Results” and Section 4 to “Discussion”. Both sections should be aggregated in one single section named “Results and Discussion”. Regarding section 3, To analyze the results statistically, the authors used different statistical tools: a series of Mann-Whitney U-tests, a mixed model ANCOVA with an unstructured covariance type to allow for the possible time dependence of observations within each subject, and a model based on Kuehl (2000). The methods are not sufficiently grounded and some validation with prior applications is needed. Tables 3 to 5 are incomplete and poorly discussed and analyzed. The graph in Figure 4 must be extensively analyzed and related and some other graphs must be added. In terms of figures and graphics, the paper must be improved.
Section 4 (Discussion) is very incomplete. Which are the main findings? Authors refer that results presented are analyzed from a socio-demographic point of view as a way to address energy efficiency actions, however, nothing is concluded regarding this subject. The conclusions of this comparison are very limited and slight. The authors limit themselves to develop some general considerations instead of strong and meaningful conclusions. It is important to remind that findings of fact are not conclusions!
No strong and quantified conclusions were outlined. The author limited themselves to conclude that a reminder 8 hours before the critical peak period had a greater influence on energy reduction than a two-time notification at 24 hours and 2 hours before the peak and no notification at all. A scientific paper must have a strong and meaningful conclusions section, which allows highlighting the main findings the investigation allowed obtaining.
Round 2
Reviewer 3 Report
Reviewer comment 1:
The authors don't refer in the paper abstract the main achievements of this research regarding the effect of this communication strategy on changing behaviors and reducing energy consumption. They just limit themselves to conclude that the main results point out some positive effects by quantifying the results of the investigation however, the authors don't refer in the article abstract the main achievements of this research regarding the strategy to adopt to mitigate peak energy demand. The abstract must include some insights on this topic.
Authors action:
A general conclusion was now added to the abstract.
Reviewer comment 2:
The structure of the article can be improved since there are two missing sections: a framing section to point out the related works on the subject, and results and discussion section where the authors should correlate the effectiveness of feedback and reminder prompts with daily peak demand. Additionally, section 5 (conclusions) should not only establish pieces of evidence based on results but also must frame this study in the context of all related studies that have played an important role in investigating motivational strategies for reducing daily peak demand. A better organization of the paper is strongly needed.
Authors action:
The structure of the paper was considerably improved. Some new sections were now added, mainly concerning the literature review and result discussion. The conclusions section was improved.
Reviewer comment 3:
There is a missing section to point out the related works on the subject. Section 2 refers to Materials and Methods. Section 2 must focus on a detailed literature review including related works. The authors didn't include in the research a specific section for “Literature review”. The related works are presented in Section 1 (Introduction). The number of references is short (38) and is developed without relating the quoted studies to each other. It is convenient to reduce the explanatory content of each reference plus research related to the subject of the research. Some new references should be added in a specific Section called “Literature review”, “Related works” or “State of the Art”. All new references should be related to the analyzed subject and it would be appreciated to quote some articles published in the Journal Buildings of MDPI regarding the investigation subject.
Authors action:
A new state-of-the-art section is now available, however, the number of references is definitely reduced – only 32-.
Reviewer comment 4:
In section 1 (Introduction) authors don´t refer adequately to the purpose of the research. The structure of the article is not presented. How many Sections and its name? What is the content presented in each Section? In short, how is the article organized? On the other hand, they refer that they have tried to improve energy peak demand by implementing a communication strategy. How was this improvement measured?
Authors action:
The aims of this research are now specified in the Introduction section. The organization of the paper is outlined in section 1.
Reviewer comment 5:
Section 2 refers to Materials and Methods. Figures 2 and 3 must be included in an annex to the paper text. The procedures described in Section 2 must be part of a strategy to improve communication regarding energy peak demand improvement. Therefore, the implemented methods should be better described as part of a strategy to achieve a higher purpose – reduce energy consumption and mitigate CO2 emissions-. A schematic or drawing of the general arrangement of all implemented methodology would be welcome.
Authors action:
The authors didn´t attend to the reviewers´ suggestions on this topic.
Reviewer comment 6:
Section 3 refers to “Results” and Section 4 to “Discussion”. Both sections should be aggregated in one single section named “Results and Discussion”. Regarding section 3, To analyze the results statistically, the authors used different statistical tools: a series of Mann-Whitney U-tests, a mixed model ANCOVA with an unstructured covariance type to allow for the possible time dependence of observations within each subject, and a model based on Kuehl (2000). The methods are not sufficiently grounded and some validation with prior applications is needed. Tables 3 to 5 are incomplete and poorly discussed and analyzed. The graph in Figure 4 must be extensively analyzed and related and some other graphs must be added. In terms of figures and graphics, the paper must be improved.
Authors action:
The authors didn´t attend to the reviewers´ suggestions on this topic. The quality of the figures and graphics should be considerably improved. Some images are blurry: figures 2 and 3, for instance.
Reviewer comment 7:
Section 4 (Discussion) is very incomplete. Which are the main findings? Authors refer that results presented are analyzed from a socio-demographic point of view as a way to address energy efficiency actions, however, nothing is concluded regarding this subject. The conclusions of this comparison are very limited and slight. The authors limit themselves to develop some general considerations instead of strong and meaningful conclusions. It is important to remind that findings of fact are not conclusions!
Authors action:
Some improvements have been done to the new results discussion section. However, the authors must work much more on the interpretation of the result.
Reviewer comment 7:
No strong and quantified conclusions were outlined. The author limited themselves to conclude that a reminder 8 hours before the critical peak period had a greater influence on energy reduction than a two-time notification at 24 hours and 2 hours before the peak and no notification at all. A scientific paper must have a strong and meaningful conclusions section, which allows highlighting the main findings the investigation allowed obtaining.
Authors action:
The conclusions section is now improved. Additionally, some strategic actions on energy reduction should be highlighted.
